# The Menu Served in Canadian Penitentiaries: A Nutritional Analysis

**DOI:** 10.3390/nu14163400

**Published:** 2022-08-18

**Authors:** Claire Johnson, Charlotte Labbé, Anne Lachance, Caroline P. LeBlanc

**Affiliations:** 1École des Hautes Études Publiques, Université de Moncton, Moncton, NB E1A 3E9, Canada; 2École des Sciences des Aliments, de Nutrition et d’Études Familiales, Université de Moncton, Moncton, NB E1A 3E9, Canada

**Keywords:** prison menu, incarceration, prison population, nutritional analysis, dietary reference intake

## Abstract

The food served in Canadian penitentiaries was scrutinized following food service reform where Correctional Service Canada (CSC) created a standardized menu to feed incarcerated male individuals. Food in prison is a complex issue because penitentiaries are responsible for providing adequate nutrition to the prison population, who are vulnerable to poor health outcomes but are often seen as undeserving. This study aimed to analyse the national menu served in Canadian penitentiaries, in order to compare them with Dietary Reference Intakes (DRIs) for male adults and the internal nutritional assessment reported by CSC. The goal was to verify if the menu served was adequate and to validate CSC’s nutritional assessment. The diet analysis software Nutrific^R^ was used to analyse the 4-week cycle menu. Both analyses were within range for DRIs for most nutrients. However, some nutrients were not within target. The sodium content (3404.2 mg) was higher than the Tolerable Upper Intake Levels (UL) of 2300 mg, the ω-6 (linolenic acid) content (10.8 g) was below the AI of 14 g, and the vitamin D content (16.2 μg) was below the target of 20 μg for individuals older than 70 years. When these outliers were analysed in-depth, the menu offering was consistent with the eating habits of non-incarcerated individuals. Based on this nutritional analysis and interpretation of the results in light of the complex nature of prison food, this study concludes that CSC meets its obligation to provide a nutritionally adequate menu offering to the general population during incarceration.

## 1. Introduction

In 2012, food served during incarceration in Canada was highly scrutinized following food service reform among Canadian penitentiaries [1]. The reform was called “Food service modernization” and was a part of the larger deficit reduction plan of the Federal government of Canada [1]. At the time, Correctional Service Canada (CSC) proposed a national menu to standardize the food offered during incarceration in federally run penitentiaries across Canada, thereby ensuring nutritional standards are provided to all incarcerated individuals. A national menu could reduce spending by purchasing food items in bulk and improving purchasing power. Before this reform and the national menu, separate penitentiaries had more leeway. They could serve the food of their choosing, which led to a convoluted mismatch in food offerings across Canada.

Following the changes proposed in the national menu, the importance of food served in Canadian penitentiaries was highlighted by multiple complaints, media articles, and other discussions around prison food [2,3,4]. The questions about prison food are important and complex for several reasons. The prison population is considered vulnerable, with multiple health disparities and a web of negative health profiles [5]. The food offering and the food environment within the penitentiary influence weight gain during incarceration [6,7,8]. In fact, people who are incarcerated are more likely to be overweight or obese than the general Canadian population [7], ultimately putting them at a disproportionately higher risk of developing obesity-related comorbidities during their incarceration [9].

Not only is the prison population vulnerable, but others often see incarcerated individuals as undeserving of support. The literature on deservingness has shown that citizens of advanced democracies are more likely to support public spending when it is directed at people whose situation of hardship is seen to be out of their control and who they see as similar to themselves [10]. This contrasts sharply with the way the carceral population is portrayed in Canada. In public discourse, particularly that of politicians who subscribe to the “tough on crime” approach to justice, people who are incarcerated are seen as inherently different from “ordinary law-abiding Canadians” [11]. As such, they should be “held accountable” for their actions through harsher punishments [12]. When compared to groups perceived to be more deserving, such as the elderly or families with children, incarcerated people are seen as undeserving of public spending [12]. The deservingness literature shows that programs that focus on groups seen as less deserving by the public are more likely to suffer budgetary cuts [10]. This was the case in carceral policy with the “Food modernization program”, which was created as a means to fight public deficit [1]. As the lack of funding may affect the quality of food in prisons, it is vital to assess whether the menus, constructed with as little funding as five dollars per person a day [3,4], are sufficiently nutritious and meet the needs of this specific population.

Correctional Service Canada is the Federal agency responsible for all incarcerated individuals who receive a sentence longer than two years in Canada [13]. Therefore, CSC is accountable for adequately feeding them and must adhere to government-issued nutritional guidelines [14,15,16,17,18]. The standards for nutritional guidelines in Canada are Dietary Reference Intakes (DRIs) from Health Canada [16]. The DRIs are a set of scientifically based nutrient reference values for a healthy population and are used by dietitians to assess nutritional intake in the Canadian population.

Information on diets and nutrient intake during incarceration is limited worldwide and non-existent in Canada to our knowledge. Two American studies in the last decade have shown that the menu served in prisons typically has certain nutrients that are either in excess or in inadequate quantities for the prison population [19,20]. Recently a Polish study reported that approximately 90% of individuals exhibited a low value of the Pro-Healthy Diet Index during incarceration using the KOMPan questionnaire [21]. Given the lack of research on the nutritional content of food served during incarceration in Canadian penitentiaries and the reported mistrust towards CSC’s ability to adequately feed individuals during incarceration [3,22], this study aimed to analyse the national menu served to individuals in Canadian penitentiaries and to compare our findings with the DRIs for male adults [16]. As a secondary goal, this study also aimed to compare the results of the nutritional analysis to those reported by CSC during their internal nutritional assessment.

Table 1 below presents the sociodemographic information for a sample of incarcerated individuals to provide a general idea of the prison population in Canadian penitentiaries [7].

## 2. Materials and Methods

Correctional Service Canada’s national headquarters in Ottawa provided a copy of their 4-week cycle (or 28-day) national menu and the results of their internal nutritional assessment through an access to information and privacy request [23], which included energy, fat, protein, carbohydrate, fibre, sodium, potassium, calcium, and vitamin C. The national menu is used to feed male individuals incarcerated in 31 institutions across all five geographical regions of Canada (Atlantic, Québec, Ontario, Prairie and Pacific regions) [24]. These penitentiaries house approximately 13,500 individuals [24], adding up to over 40,000 meals served daily (breakfast, lunch and dinner).

In this study, the DRIs were used to assess the adequacy of the national menu. When there is sufficient evidence, DRIs typically suggest the “Recommended Dietary Allowance” (RDA) as a reference goal for usual intake by healthy individuals [16]. However, when there is insufficient evidence to support RDA, “Adequate Intake” (AI) is used as the goal for usual intake for individuals. The DRIs also include the “Acceptable Macronutrient Distribution Ranges” (AMDR), which is a range of intake for protein, fat and carbohydrate that is associated with reduced risk of chronic disease while providing adequate intake of essential nutrients [16]. In comparison, the “Tolerable Upper Intake Level (UL)” is the highest average daily nutrient intake likely to pose no risk of health effects to most individuals [16]. Correctional Service Canada’s menu planning guidelines follow the AMDR for adults to establish menu targets for protein, fat and carbohydrates and RDA or AI to set targets for vitamins, minerals and fibre [15,16]. The DRI equation for males 19 years and older was used to estimate the prison population’s energy requirement (EER) (kcal/day). Since 45% of the prison population is obese [7], the adjusted body weight was used to assess the energy needs. That means that 25% of the difference between actual body weight and ideal body weight is added to ideal body weight to assess nutritional needs [25]. With this approach, the dietary needs were based on an adjusted body weight standard of 82 kg (instead of the ideal body weight of 78 kg). The average height and age used in calculations were based on a recent publication of our research team: 1.76 m for average height and 43 years for average age [26]. The average physical activity level (PAL) of 1.11 (low active or typical living activities plus 30–60 min of moderate activity) was used to calculate energy needs for the prison population [16,26,27].

The Diet Analysis software Nutrific^R^ from Université Laval (Quebec, Canada) was used to analyse the menu [28]. Nutrific^R^ is a web application that uses the data from the Canadian nutrient files [29]. The menu items for each cycle day from CSC’s national menu were inputted into Nutrific^R^ separately by two research assistants with backgrounds in dietetics (one registered dietitian and a fourth-year student in dietetics). The nutritional data for each day from Nutrific^R^ was analysed to provide daily totals for each nutrient. Once completed, the research assistants compared their findings to reduce the chance of errors; when a discrepancy occurred, they investigated the source of the error. When this exercise did not reveal the source of the error, they asked the professor supervising the study to investigate if the error was not apparent to them. These errors only occurred twice, representing an error rate of 2.8%. Once the nutritional analysis was completed, we were confident it did not contain significant errors. The data were then inputted into Statistical Package for Social Sciences (SPSS) Statistics version 28 [30] for descriptive statistical analysis (mean and standard deviation). The data were again reviewed using SPSS to verify mistakes or outliers in the dataset. Once completed, the data were used to compare means between the three analyses (research assistants 1 and 2 and CSC’s analysis) with a multivariate analysis of variance (MANOVA). In preparation for the MANOVA testing, preliminary analyses were performed to verify outliers in the data, normality of the distributions, linearity between the dependant variables and homogeneity of variances. The goal of the statistical analysis was to determine if there were significant differences between our analysis and the one conducted at CSC. All significance levels were set at 0.005.

## 3. Results

Table 2 presents this study’s nutritional analysis of the national menu compared to the internal nutritional assessment obtained from CSC. The third column presents current Canadian DRIs. When broken down, both nutritional analyses were within range for DRIs for most nutrients since the menu content was higher than the RDA and was also below the UL. This was the case for calcium, iron, vitamins A, C, D, E, B6, niacin, folate, phosphorus, magnesium, copper and manganese. For other nutrients, the menu content was higher than RDA, but there was no UL set by Health Canada. This was the case for fibre, cholesterol, vitamins K, B1, B12 and pantothenic acid. However, the national menu’s average sodium content (3404.2 mg) was higher than the Adequate Intake (AI) of 1500 mg and was also higher than the UL of 2300 mg. The ω-6 (linolenic acid) content (10.8 g) was below the AI of 14 g. The vitamin D content (16.2 μg) of the menu meets the AI (15 μg/day) for males aged 18 to 70 years. However, the vitamin D content is below the target for individuals older than 70 years whose AI is 20 μg per day.

Overall, there were no statistically significant differences between the analysis of variance between the first and second research assistants (*p*-values ranging between 0.066 and 0.950). Therefore, the data from the first research assistant were used to compare our results (means) to the results (means) provided by CSC. Overall, our nutritional analysis was similar to the one conducted by CSC, as shown in Table 2. There was, however, a significant difference (*p* < 0.001) between the amount of calcium found during our analysis (1673.6 mg) and the amount from the analysis by CSC (1410.1 mg). Moreover, there was a significant difference found between the average protein contents (*p* = 0.002) between both analyses. The analysis for this study found the national menu to be higher in protein (129.1 g) vs. the amount reported by CSC (118.5 g).

## 4. Discussion

This study’s main finding is that, overall, CSC’s national menu or the food offering in Canadian penitentiaries is nutritionally adequate. The national menu meets most nutritional requirements set by Health Canada when broken down to the nutrient level. Typically, when the nutrient content of the menu was higher than the RDA, it was also below the UL. This is the case for calcium, iron, vitamins A, C, D, E, B6, niacin, folate, phosphorus, magnesium, copper and manganese. In other cases, when the menu content was higher than the RDA, there was no UL set by Health Canada. This typically suggests there is no known effect on health if intake is high from food, as was the case for fibre, vitamins K, B1, B12 and pantothenic acid [16,31]. Three nutrients (sodium, ω-6 (linolenic acid) and vitamin D) were not within target with current DRIs and will be discussed further.

The sodium content of the national menu is higher than the AI of 1200–1500 mg and the UL of 2300 mg. Sodium provisions are a concern in most correctional facilities, as the offering is usually above recommendations. In one American study, the average sodium content from the menu was 4500 mg [19], whereas in another, the menu provided amounts (3400 mg) similar to our study [20]. Generally speaking, most Canadians (incarcerated or not) consume more sodium than is recommended [32], which makes providing a menu with less than 2300 mg of sodium difficult and unpalatable. Since 2010, a reduction in the consumption of sodium has been a priority for Health Canada as it implemented the “Sodium reduction strategy” [33]. The Canadian community health data from 2015 found a decrease in average sodium consumption in male adults (aged 19–70 years) since 2010 [32]. The average intake for Canadians (male and female) was 3400 mg in 2010 [34], which is similar to the amount found in our study (3404.2 mg). Furthermore, the average consumption is based on self-reported data, which are likely an underestimation of intake. In fact, when using recovery biomarkers to compare sodium intake, a recent study found that sodium intake was underestimated by approximately 30% [34]. Based on those findings, that means sodium intake is likely closer to 4000 mg in that study [34].

The issue around sodium is complex since there is evidence that during incarceration, people rely heavily on food from the commissary store, which is basically a convenience store within the penitentiary [20,35,36,37]. Commissary stores typically sell junk food [37,38] and the food purchased there generally is high in sodium [39]. The food available in the Canadian prison environments comes from two sources: food provided by food services and the commissary store [6]. There is a delicate balance between the two, and if the food offered on the menu is deemed unpalatable, then people will turn to the commissary store for food. This may be more harmful to their health than eating from a menu that contains more sodium than recommended [6]. Given that prison food is part of a closed system (this means only food from food services and the commissary store is permitted for individuals living in Canadian penitentiaries), a more liberal approach to sodium to increase acceptability may be justified and better for long-term health.

According to our analysis, the daily average for ω-6 (linolenic acid) content of the national menu (10.8 g) was below the current Canadian AI of 14–17 g. The Canadian community health survey results show that all Canadian adults, regardless of age and gender, have median intakes of linoleic acid below their AIs. Despite this finding, ω-6 deficiency is basically non-existent in Canada [40]. To interpret this finding, it is important to understand how the AI for this nutrient was established. The AI for linoleic acid is based on observed intake from the US. Canadian food supply may provide less linoleic acid due to the preferential use of canola oil rather than soybean oil. Therefore, an assessment of adults’ linoleic acid intake using an AI based on Canadian data may have had a different outcome [40]. Our analysis revealed a well above daily average for ω-3 alpha-linolenic acid (3.2 g vs. 1.6 g for the AI). This mirrors Canadian adult consumption [40]. However, it seems that the ratio of ω-6:ω-3 might be more conclusive than individual content or intake as it is an indicator of the balance between ω-6 and ω-3 acids in the diet [40]. The target ω-6:ω-3 ratio for optimal human health should be between 1:1 and 2:1 [41]. However, the ratio in Canadian adults’ diets varies between 7:1 and 8:1, depending on the age and sex [40]. The national menu provides a ω-6:ω-3 acid ratio of 3.4, which is far superior to the Canadian average.

Adult men need 15–20 µg of vitamin D daily as per the RDA, depending on age [16]. The national menu revealed average daily content of 16.4 µg. While this amount is sufficient for adult men up to 70 years of age, those above 70 years of age need more (20 µg daily) [16]. Current Canadian guidelines advise all Canadians who are 51 years of age and older to take a daily supplement containing 10 µg (or 400 IU) of vitamin D while maintaining a diet containing foods with vitamin D as part of healthy eating [42]. As shown in Table 1 of this paper, only a small proportion (3%) of individuals incarcerated in Canadian penitentiaries are above 70 years of age and would require a supplement (or extra vitamin D from food) to meet RDA targets. This could likely be managed through a nutrition management program without necessarily changing the menu for the general population [43,44].

Various methods are used to assess a prison population’s nutritional needs. Some studies assessing prison menus used different strategies to determine the dietary needs of incarcerated individuals. For example, Collins and Thompson used a reference body mass index (BMI) of 22 to calculate nutritional needs for males incarcerated in the United States [20]. In contrast, another study used a standard reference male based on the average height and weight of non-incarcerated adults in the United States because there were no available data on the anthropometrics of incarcerated individuals [19]. For the nutritional needs assessment of this study, we could have used the actual mean weight for the prison population in Canadian penitentiaries, which is 93.3 kg, and the mean height of 1.76 m, because the data were available [26]. However, with actual mean weight and height, the mean BMI is in the obese range (BMI = 30). Using them could overestimate the caloric needs of the incarcerated males and potentially increase further weight gain [7]. Whereas, using ideal weight to calculate caloric needs could underestimate nutritional needs and may lead individuals to use commissary foods as a supplement to the national menu, which may also lead to further weight gain [6,35,38,45]. We opted for the adjusted body weight method, where 25% of the difference between actual and ideal body weight is added to ideal body weight to assess nutritional needs [25] as a more balanced approach. Ultimately, the dietary needs were based on an adjusted body weight standard of 82 kg (instead of the ideal body weight of 78 kg). This more nuanced approach is needed because of the complex nature of the environment and the unique food dynamic of the closed food system in Canadian penitentiaries.

The nutritional analysis results for this study did not reveal any inadequate nutrients that could cause nutritional deficiencies in healthy male adults if following this diet long term. CSC’s nutritional analysis was more superficial (only analysing nine nutrients) than the one performed for this study, where 35 nutrients were analysed. Based on this study’s findings, which provide a much more in-depth nutritional analysis that includes types of fat, more vitamins and minerals, CSC meets its obligation to provide a nutritionally adequate menu offering during incarceration [18,46]. In addition, the results of this study confirm the validity of the nutritional analysis performed by CSC.

Two nutrients (protein and calcium) had statistically significant variations between our nutritional analysis and CSC’s internal nutritional assessment. The difference found for both nutrients can be explained by the protein and calcium content in milk (316 mg of calcium and 8.7 g of protein per 250 mL) [29] vs. content in the *skim milk plus 100 powder* (240 mg of calcium and 5 g of protein per 250 mL) served as part of the national menu [47]. The statistical analysis of this study and CSC’s internal nutritional assessment found a significant discrepancy for calcium. Correctional Service Canada found 1410.07 mg (*p* < 0.001 * when compared to 1673.6 mg). A similar trend was observed with the protein content, where the analysis for this study found 129.1 g of protein, whereas the analysis from CSC found 118.5 g of protein (*p* = 0.002 *). Three cups (750 mL) of dairy beverage or *skim milk plus 100 powder* is served daily on the national menu, which provides 720 mg of calcium and 15 g of protein. However, this product is not available in the Canadian Nutrient File [29]; therefore, skim milk was used for the nutritional analysis for this study. Three cups (750 mL) of skim milk provide 948 mg of calcium and 26.1 g of protein. The difference in calcium and protein content between both products (228 mg of calcium and 11.1 g of protein) explains the difference in calcium between this study’s findings and CSC’s nutritional assessment. Even with the lower calcium and protein content of the *skim milk plus 100 powder,* the amount of calcium and protein provided on the national menu still exceeds the RDA for calcium (1000–1200 mg) and AMDR for protein (68–240 g) [16]. The primary reason for switching from regular fluid milk to *skim milk plus 100 powder* was for cost savings as part of the food service modernization initiative in response to the deficit reduction plan [1,48].

Budgetary concerns are a priority when feeding the prison population in Canada. The word “per diem” is regularly used in penitentiaries to describe the daily allowance (or a specific amount of money) provided for each incarcerated individual’s food cost. In Canada, the per diem to feed people during incarceration was around CAD 5.41 when this study was conducted [3,4]. These low figures are explained by the reluctance of recent Canadian governments to increase funding for the welfare of the prison population. This is consistent with the literature on deservingness, which shows that programs targeting people seen as undeserving of state support are more likely to suffer budgetary cuts [10]. As mentioned above, politicians subscribing to the “tough on crime” strategy have portrayed incarcerated people as undeserving of state funding, describing them as inherently different from other Canadians, and arguing that “holding criminals accountable” should be achieved through harsher sentences and stricter detention conditions [11,12]. In surveys conducted on this issue, the Canadian public has shown that it believes that sentences are not severe enough [49]. Under the leadership of Prime Minister Stephen Harper, carceral policies were altered to fit this vision, and the government implemented severe cuts to prison budgets, including monies used to procure food [1]. However, despite these budget cuts, CSC has managed to provide adequate nutrition to the general prison population using the national menu.

This punitive approach is less ingrained in policy in Canada than it is in the United States, where the amount of money to feed individuals during incarceration is much lower. In the United States, the national food cost average is around USD 3.32 (CAD 4.26) [20,50]. However, there are multiple reports of food costs as low as USD 1.13 (CAD 1.45) in certain American correctional facilities [20,51]. A study out of the United States argues that the low cost of food is related to the privatization of correctional facilities that prioritizes profits over the health and well-being of individuals during incarceration and under the “care” of the state [52]. The authors raise ethical concerns about the rights of the vulnerable population, who will often be at the mercy of prison authorities regarding their food offerings with very little recourse [52]. Certain correctional facilities in the US have daily menu offerings as low as 1700 calories [52]. That appears to be an outlier since most American correctional facilities provide an average of 2600 calories daily [19,20]. The calories provided are still considered low. Generally, a menu providing 2800 calories is deemed adequate [52] for adult males during incarceration. The findings of this study estimated that the national menu (in Canadian penitentiaries) offers an average of 2824 calories daily.

The findings of this study support that, at the very least, the national menu provides adequate nutrition and obtains more funding compared to the food offerings and budget in certain American prisons [19,20,52]. However, as mentioned above, milk was replaced by *skim milk plus 100 powder* in Canadian menus to cut costs, which may negatively affect the calcium intake of incarcerated Canadians. This is especially true if people do not consume the powdered milk because they dislike it or it is perceived as punishment [36]. There were other motivations, beyond cost savings, for choosing powdered milk. For instance, powdered milk has a longer shelf-life which reduces the risk of wasted product, and powdered milk reduces paper waste at the institutional level compared to the individual milk cartons used before the national menu [47]. Food in prison goes far beyond the role of nourishment and physical health [53,54]. Food can be a source of pleasure [36], act as a substitute currency [6,55], can be used as punishment or power [36] and as a source of autonomy [35,36]. This study focused on the nutritional adequacy of the national menu or the food offered in Canadian penitentiaries without assessing the other roles prison food can play. These roles could be examined in further publications.

This study is an essential first step in assessing if incarcerated individuals are adequately fed. The national menu content is necessary and must be nutritionally adequate. However, a significant limitation of this study is that the nutritional analysis is only a part of the story. Given CSC’s complex production and distribution system, this study does not guarantee that each institution follows the national menu adequately. There is some evidence of inconsistencies in following the prescribed menu and providing the portion sizes outlined in the national menu [48]. In addition, the food offered on the national menu does not guarantee adequate food intake per se [45]. There is evidence that regardless of the menu offering, some people rely heavily on food from the commissary store, which is typically considered unhealthy and is related to weight gain during incarceration [6,35,52]. Although limited, incarcerated individuals have some control over making their own food choices. They are not forced to consume the food served to them [14]. In addition to the food provided by food services, they can purchase extra food from the prison commissary store, usually once a week (with their funds). Data from one of the prisons in Australia show that up to 30.5% of the energy consumed comes from food products that were purchased by the individuals themselves [45]. A study conducted on a group of 307 male participants in an addiction treatment program in an American prison reported that only 6.7% of them exhibited healthy diet choices [56]. In Canadian penitentiaries, therapeutic, cultural, religious and other potential food accommodations impact food offerings [18,43,57,58]. Further research is needed to determine food intake and the nutritional quality of all the food consumed (including food from the commissary store and other diets) during incarceration in Canadian penitentiaries.

For continued improvement of the menu offered to incarcerated individuals in Canadian penitentiaries, it could be helpful to identify the highest sodium food served (ham, hot dogs, baked beans, teriyaki chicken, tuna salad sandwich, whole wheat bread) and assess appreciation for those foods. If a high-sodium food is also unpopular, that food could be removed or replaced with a lower-sodium option. The effort to reduce the sodium content of the menu must be slow and gradual to improve the acceptability of the changes to the menu. In addition, to ensure adequate vitamin D consumption for individuals over 70 years, a fourth glass of milk or a vitamin D supplement may be offered.

## 5. Conclusions

Overall, the menu served during incarceration in Canadian penitentiaries is nutritionally adequate, especially given the complex nature of feeding the prison population with a closed system, where individuals have limited food choices (CSC menu offerings and commissary store). It is a delicate balance between the menu and the commissary store, because if the menu is deemed undesirable or unpalatable, there is evidence that individuals may purchase more non-nutritious foods from the commissary store. In the long run, the food purchased from the commissary store may lead to more health problems for the already vulnerable prison population. Although some nutrients (sodium, ω-6 (linolenic acid) and vitamin D) are technically not meeting targets, some of them are deemed acceptable (for sodium and ω-6 or linolenic acid) when interpreted in light of the complex nature of prison and when compared to the eating habits of non-incarcerated Canadians. Minor therapeutic intervention (for vitamin D) may be helpful for individuals older than 70 years, but this does not impede the value of the national menu for the general prison population. In addition, the internal nutritional assessment performed by CSC is validated since there were few significant variations when comparing results from this study to their internal nutritional assessment. The only two discrepancies (calcium and protein) were due to the nutritional difference between powdered milk and fluid milk. Moving forward, it could be beneficial to study actual food intake during incarceration in Canadian penitentiaries since there is evidence that suggests it is likely different than the nutritional value of the food served on the national menu. This study is the beginning of a broader conversation regarding public funding, perceived deservingness and the collective responsibility to protect the health of one of the most vulnerable segments of the Canadian population.

## Figures and Tables

**Table 1 nutrients-14-03400-t001:** Sociodemographic factors from a study sample of incarcerated individuals in Canada (N = 1420).

Factors		All N (%) 1420 (100)
Sex	Male Female	1276 (89.9) 144 (10.1)
Age	18 ≤ 24 years ≥25 ≤ 34 years ≥35 ≤ 44 years ≥45 ≤ 64 years ≥65 ≤ 70 years >70 years	104 (7.3) 389 (27.4) 315 (22.2) 504 (35.5) 64 (4.5) 44 (3.1)
Ethnicity	Caucasian Black Indigenous Other	904 (63.7) 203 (14.3) 214 (15.1) 99 (7.0)
Security level	Maximum Medium Minimum	348 (24.5) 781 (55.0) 291 (20.5)
Length of incarceration at follow-up	≤18 months >18 months ≤ 5 years >5 years	553 (38.9) 458 (32.3) 409 (28.8)
Sentence total	2 ≤ 3 years >3 ≤ 5 years >5 ≤ 25 years >25 years	285 (20.1) 286 (20.1) 365 (25.7) 484 (34.1)

**Table 2 nutrients-14-03400-t002:** Mean content per nutrient of our nutritional analysis compared to CSC’s internal analysis, CSC nutritional targets and current Canadian DRIs.

	Our Nutritional Analysis	CSC Nutritional Analysis	DRIs
Calories (kcal)	2824	2861	2753 ^c^
Fat (g)	83.7	81.8	61 (20%) ^d^ 107 (35%)
Saturated fat (g)	20.3	NA	As low as possible ^e^
Trans fat (g)	0.8	NA	As low as possible ^e^
Polyunsaturated fat (g)	19.2	NA	NA
ω-6 linolenic acid (g)	10.8	NA	14 ^f^ 17
ω-3 alpha-linolenic acid (g)	3.2	NA	1.6 ^f^
Monounsaturated fat (g)	34.9	NA	NA
Sugar (g)	161.8	NA	<172 (<25%)
Carbohydrate (g)	402.3	420.5	310 (45%) ^d^ 447 (65%)
Fibre (g)	38.0	39.1	30–38 ^f^
Protein (g)	129.1	**118.5 ^a^**	68 (10%) ^d^ 240 (35%)
Cholesterol (mg)	324.3	NA	As low as possible ^e^
Sodium (mg)	3404.2	3490.1	1200–1500 ^f^ 2300 ^g^
Potassium (mg)	5351.6	5383.9	4700 ^f^
Calcium (mg)	1673.6	**1410.1 ^b^**	1000–1200 ^h^
Iron (mg)	21.1	NA	8 ^h^
Vitamin A (μg)	2008.5	NA	900 ^h^
Vitamin C (mg)	154.0	147.9	90 ^h^
Vitamin D (μg)	16.2	NA	15–20 ^h^
Vitamin E (mg)	15.4	NA	15 ^h^
Vitamin K (μg)	185.4	NA	120 ^h^
Thiamine (mg)	3.2	NA	1.2 ^h^
Riboflavin (mg)	3.7	NA	1.3 ^h^
Niacin (mg)	58.2	NA	16 ^h^
Vitamin B6 (mg)	3.0	NA	1.3–1.7 ^h^
Folate (μg)	624.5	NA	400 ^h^
Vitamin B12 (μg)	8.3	NA	2.4 ^h^
Pantothenic acid (mg)	10.7	NA	5 ^f^
Phosphorus (mg)	2402.2	NA	700 ^h^
Magnesium (mg)	549.9	NA	400–420 ^h^
Zinc (mg)	18.0	NA	11 ^h^
Selenium (μg)	179.2	NA	55 ^h^
Copper (mg)	2.0	NA	0.9 ^h^
Manganese (mg)	7.8	NA	2.3 ^f^

*p*-value is the result of a multivariate analysis of variance (MANOVA) between the nutritional analysis for this study and the CSC internal analysis. **^a^ *p*-value < 0.005, ^b^ *p*-value < 0.001,**^c^ EER = 662 − (9.53 × age) + PAL ((15.91 × weight) + (539.6 × height)), ^d^ AMDR, ^e^ There is no UL for saturated fats. The recommendation in the Dietary Reference Intakes (DRIs) is simply “As low as possible while consuming a nutritionally adequate diet”, ^f^ AI, ^g^ UL, ^h^ RDA. NA, not available.

## Data Availability

The dataset generated during this study is not publicly available because the Université de Moncton does not have a place to store datasets.

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
