# Peer review of "The Menu Served in Canadian Penitentiaries: A Nutritional Analysis"

_nutrients, 2022, doi:10.3390/nu14163400_

Round 1

Reviewer 1 Report

The topic is very interesting and important.  It is very important to provide adequate nutrients to incarcerated individuals. However, I have the following comments/ suggestions:

a) Table 1 indicates that the highest age group is ³ 45 < 64 (35.5%) and the lowest age group is the individuals > 70 years old. Then why authors are referring to the nutrients need of individuals older than 70 years? See the paragraph below:

“The sodium content (3404.2 mg) was higher than the Upper Intake Levels (UL) of 2300 mg, the ω-6 (linolenic acid) content (10.8 g) was below the AI 20 of 14 g, and the vitamin D content (16.2 mg) was below the target of 20 mg for individuals older than 70 years”.                                            May be the focus should be on the age group of 45-64 years old.

b) The men population (89.9%) is much higher than women (10.1%). Are there any data on the nutritional adequacy for the incarcerated women?

c) The average dietary content for general public is often has similar deficiencies and excess (Na), as it was concluded by the authors. 

It is encouraging to see that only issue is with 3 nutrients and not more.

Author Response

The topic is very interesting and important. It is very important to provide adequate nutrients to incarcerated individuals. However, I have the following comments/ suggestions:

a) Table 1 indicates that the highest age group is ³ 45 < 64 (35.5%) and the lowest age group is the individuals > 70 years old. Then why authors are referring to the nutrients need of individuals older than 70 years? See the paragraph below:

"The sodium content (3404.2 mg) was higher than the Upper Intake Levels (UL) of 2300 mg, the ω-6 (linolenic acid) content (10.8 g) was below the AI 20 of 14 g, and the vitamin D content (16.2 mg) was below the target of 20 mg for individuals older than 70 years".                                            May be the focus should be on the age group of 45-64 years old. 

Response: We mentioned individuals who are older than 70 years because the vitamin D content of the menu is inadequate for them. Their RDA is 20 mg/day. Consequently, it is less important to focus on the more populous group of individuals aged 45-64 years because menu provides enough vitamin D for their RDA (15 mg/day) for vitamin D are met. We provided more information in the article to clarify this. See lines 168-170.

b) The men population (89.9%) is much higher than women (10.1%). Are there any data on the nutritional adequacy for the incarcerated women? 

Response: No, women incarcerated in Canadian penitentiaries do not follow the same menu provided to men. That menu has not yet been analyzed. It would be essential to analyze the women's menu during incarceration in future research.

c) The averagedietary content for general public is often has similar deficiencies and excess (Na), as it was concluded by the authors.  

It is encouraging to see that only issue is with 3 nutrients and not more. 

Response: Thank you for this comment. We agree!

Reviewer 2 Report

Johnson et al analyzed the nutritional components of the menu in Canadian penitentiaries, which was ignored previously. Generally, this study is of significance but should consider conducting a survey that focuses on the eating habits of incarcerated individuals. In addition, suggestions that can be considered to improve the current menu should be discussed.

Author Response

Johnson et al analyzed the nutritional components of the menu in Canadian penitentiaries, which was ignored previously. Generally, this study is of significance but should consider conducting a survey that focuses on the eating habits of incarcerated individuals.

Response: Yes, we agree a survey of food intake among incarcerated individuals would certainly be important information to add to the findings of this study. However, as a first step, doing a nutritional analysis of the food offered on the menu is also essential because it allows us to know if the food provided is nutritionally adequate. The food offered is only one side of the story and is probably different from the food consumed, but we still think this menu analysis in this study is a crucial first step.

In addition, suggestions that can be considered to improve the current menu should be discussed.

Response: We added a paragraph for suggestions on how to improve the menu. See lines 394-401. 

Round 2

Reviewer 1 Report

Thank you very much for the revisions. 

Reviewer 2 Report

The authors have addressed the reviewer's comment.